# The Strange-Metal Behavior of Cuprates

Giovanni Mirarchi [1,*], Götz Seibold [2], Carlo Di Castro [1], Marco Grilli [1,3] and Sergio Caprara [1,3,*]

1 Dipartimento di Fisica, Sapienza Università di Roma, P. le Aldo Moro 5, 00185 Roma, Italy; carlo.dicastro@roma1.infn.it (C.D.C.); marco.grilli@roma1.infn.it (M.G.)
2 Institut für Physik, BTU Cottbus-Senftenberg, P.O. Box 101344, D-03013 Cottbus, Germany; seibold@b-tu.de
3 ISC-CNR, Unità di Roma Sapienza, P. le Aldo Moro 5, 00185 Roma, Italy
* Correspondence: giovanni.mirarchi@uniroma1.it (G.M.); sergio.caprara@uniroma1.it (S.C.)

**Abstract:** Recent resonant X-ray scattering experiments on cuprates allowed to identify a new kind of collective excitations, known as charge density fluctuations, which have finite characteristic wave vector, short correlation length and small characteristic energy. It was then shown that these fluctuations provide a microscopic scattering mechanism that accounts for the anomalous transport properties of cuprates in the so-called strange-metal phase and are a source of anomalies in the specific heat. In this work, we retrace the main steps that led us to attributing a central role to charge density fluctuations in the strange-metal phase of cuprates, discuss the state of the art on the issue and provide an in-depth analysis of the contribution of charge density fluctuations to the specific heat.

**Keywords:** high-temperature superconductors; cuprates; charge density fluctuations; strange metal; dynamical quantum criticality

## 1. Introduction

Cuprates constitute a class of materials with tetragonal symmetry made of $CuO_2$ layers alternating with layers of oxygen and other metals [1,2]. These latter layers can be chemically doped and act as charge reservoirs for the $CuO_2$ planes. Both hole and electron doping are possible; in this paper, we focus on the most common hole doped cuprates. The cuprate era began in 1986 when J. G. Bednorz and K. A. Müller observed high-temperature superconductivity in a La-based cuprate [3,4]. Other high-temperature superconducting cuprates were identified afterward [5,6]. Superconductivity in these systems is characterized by *d*-wave symmetry, and the nature of the mediator of pairing is still debated.

Though high-temperature superconductivity is probably the most interesting characteristic of cuprates, several anomalous phenomena have been observed in their non-superconducting phase, the full understanding of which is still an open problem and is one of the major challenges of theoretical condensed matter physics. In particular, Landau's Fermi Liquid (FL) theory seems to fail in describing the behavior of the metallic state of cuprates above the superconducting critical temperature $T_c$. We quote here two features that are the hallmark of the so-called strange-metal behavior of cuprates: (i) a linear temperature dependence of the electrical resistance over an uncommonly large temperature range [7–9] that may extend from just above $T_c$ (and even below it, when superconductivity is suppressed by a magnetic field [10]) up to the highest measured temperatures and (ii) a logarithmic temperature dependence and a non-monotonic doping dependence of the specific heat, which seems to diverge at $T = 0$, for a specific value of the doping $p^*$ [11].

The main goal of this work is to provide a state-of-the-art account of the strange-metal behavior of cuprates and to discuss a theoretical scheme that can describe it. Our discussion is largely based on recent studies on this issue [12–14]. In this paper, we add a detailed description of the theory of specific heat of overdamped collective excitations. The paper is organized as follows:

- In Section 2, we briefly describe a typical phase diagram of cuprates and introduce some of the ideas that have been put forward in the attempt to find a rationale behind the various phases that occur in it and their competition/interplay.
- In Sections 3 and 4, we go through the main steps that led us to the formulation and elaboration of our model for the theoretical description of the strange-metal phase of high-temperature cuprate superconductors. In particular, an emphasis will be given to the role of the so-called charge density fluctuations that were identified in recent resonant X-ray scattering experiments on cuprates [15].
- In Sections 5 and 6, we show that the current version of our model is apt to account for the two hallmarks of the strange-metal phase of cuprates discussed above.
- In Sections 7 and 8, we will describe the technical aspects of our model in detail and provide some recent developments about the evaluation of the specific heat due to overdamped charge density fluctuations.
- Lastly, in Section 9, we make some concluding remarks and discuss the future perspectives of our work.

## 2. The Phase Diagram of Cuprates and the Quantum Critical Point Scenario

The various physical behaviors observed in cuprates are usually represented on a two-dimensional temperature vs. doping phase diagram, such as the one sketched in Figure 1. The doping level, which is commonly denoted with $p$, refers to the concentration of doped holes (electron doping is also possible but, for the sake of definiteness, we refer to the more common hole-doped case). At low doping, cuprates exhibit a fully developed antiferromagnetic order, which corresponds to a Mott insulating behavior [16–18]. Superconductivity appears underneath a dome-shaped line $T_c(p)$, for $0.05 \lesssim p \lesssim 0.25$, the maximum $T_c$ being reached at the optimal doping $p_{opt} \approx 0.19$. Samples with larger (smaller) doping are said to be overdoped (underdoped).

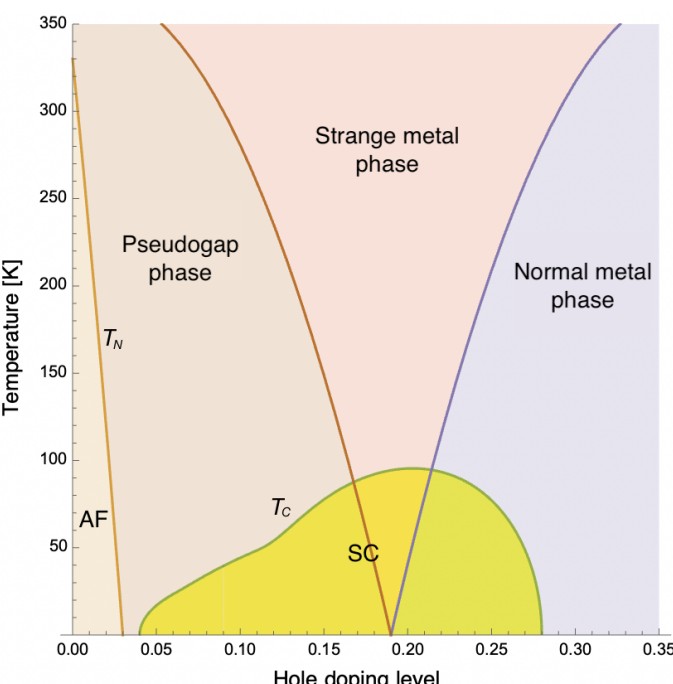

**Figure 1.** Typical temperature vs. doping phase diagram of cuprates. At temperatures $T$ larger than $T_c$, the phase diagram appears to be divided into three main regions; however, it is important to keep in mind that this division is not necessarily neat and the partitioning lines might be understood as crossover temperatures between different regimes. The regions denoted with AF and SC indicate, respectively, the antiferromagnetic and superconducting phases. These regions are sharply delimited by the Néel temperature $T_N$ and superconducting critical temperature $T_c$, respectively.

Upon doping, at (fixed) temperatures larger than the highest superconducting critical temperature $T_c$, the antiferromagnetic Mott insulator evolves into a pseudogap phase, where the system behaves as a bad metal, with several experimental probes detecting a suppression of the electron density of states at and around the Fermi energy [19–22]. The pseudogap state is a complicated realm where the main actors gradually change, spin fluctuations playing the major role at lower doping and charge fluctuations prevailing at higher doping. In most cuprates, with the noticeable exception of $La_{2-x}Sr_xCuO_4$, charge and spin degrees of freedom seem to interplay very weakly in this region of the phase diagram, their characteristic wave vectors evolve independently of each other with doping and have opposite doping dependences [23,24]. With further increasing the doping, the strange-metal phase is reached, which is characterized by a non-FL behavior in spectral [25] and transport [8,9] properties. At even larger doping, a normal metal phase, displaying a standard FL behavior, is eventually reached. The lines separating these phases do not appear as phase transition lines but rather as smooth crossover lines. It is important to point out that recent results [26,27] unambiguously show that the pseudogap develops well within the strange-metal state, and not vice versa, indicating that the strange-metal behavior must find its explanation in the absence of a pseudogap.

It was suggested long ago [28] that a rationale for the above tripartition of the metallic state (pseudogap phase/bad metal, strange metal, normal metal) can be provided by the existence of a quantum critical point (QCP), hidden underneath the superconducting dome, at a doping $p_c \approx p_{opt}$. In the standard theory of quantum critical phenomena [29], three regions exist above a QCP, a nearly ordered region (true long-range order may be suppressed by fluctuations in this region), a quantum critical region and a quantum disordered region, which in cuprates are put in correspondence with the pseudogap (bad-metal) phase, the strange-metal phase and the normal-metal phase, respectively [30]. The fact that the maximum critical temperature $T_c$ is achieved near $p_c$ is by no means accidental, as near a QCP, low-energy dynamical collective excitations can mediate a strongly doping and temperature dependent interaction among the fermion quasiparticles, which can be attractive in the $d$-wave Cooper channel [31].

However, as we shall discuss below, it became subsequently clear that the quantum disordered phase is not necessarily a strange metal, and many years were needed to identify the characteristics (from the theoretical point of view) and the nature (from the experimental point of view) of the collective excitations that are needed to explain the strange-metal behavior. The steps that we followed to arrive at the present form of our theory is traced in the forthcoming sections: the identification of charge ordering as the phenomenon related to the QCP and the discussion about its static/dynamic character; the realization that fluctuations characterized by a finite wave vector cannot be responsible for the behavior of the resistivity in the strange-metal phase when they are too close to the QCP; the way out provided by the charge density fluctuations that were identified in recent resonant X-ray scattering experiments [15]; and the understanding that, in order to give rise to an extended strange-metal behavior, charge density fluctuations must undergo a dynamical transition toward an overdamped regime at some doping $p^* > p_c$. In all our forthcoming discussion, we assume that superconducting order is suppressed, e.g., by means of a sufficiently strong magnetic field [32,33], so as to fully access the strange-metal regime, which is the focus of the present work. Although the magnetic field can also induce some changes in the fermiology of the system, we argue that charge density fluctuations are sufficiently broad in momentum space that no specific nesting condition is needed for them to act as the scatterers that are responsible for the strange-metal behavior, so our theory can withstand reasonable modifications, e.g., of the Fermi surface.

One important consequence of the existence of a dynamical instability at $p^* > p_c$ is that the wedge-like shape of the strange-metal region in the sketchy phase diagram of Figure 1 would be modified. On the doping axis, an interval $p_c < p < p^*$ would be present, with $p_c$ marking the point of an extended quantum critical behavior for the

related fluctuations, whereas the most extended strange-metal behavior would occur in correspondence of $p^*$.

## 3. Charge Ordering in Cuprates

Although the parent compounds are antiferromagnets, antiferromagnetism is very rapidly spoiled upon doping, and it seems unlikely that spin fluctuations may play a major role near optimal doping. It became gradually clear that the phenomenon related to the QCP that rules the physics of cuprates was rather related to charge ordering. One of the first examples of charge ordering in cuprates was observed in $La_{1.6-x}Nd_{0.4}Sr_xCuO_4$ [34], in the form of a static stripe-like charge order (i.e., a state characterized by a unidirectional charge density modulation), triggered by Nd co-doping, in the pseudogap phase [35,36]. Experiments showed that these stripe modulations are characterized by a strong interplay between charge and spin degrees of freedom. Nevertheless, stripe-like charge modulation seems to be a peculiarity of Nd co-doped La-based cuprates [37].

At about the same time as the first experimental observations of the stripe-like charge order, the idea that charge order might ubiquitously occur in cuprates, in the form of incommensurate charge density waves, began to emerge, mostly on the basis of theoretical considerations [28,38]. In particular, the mechanism responsible for this kind of charge order was identified in the compromise between the attraction of electron quasiparticles mediated by phonons and the Coulomb repulsion, in the presence of strong electron-electron correlations, giving rise to the so-called frustrated phase separation [39,40]. This mechanism would lead the system to a phase characterized by an uneven distribution of the electron charge over the lattice, whose spatial periodicity is not commensurate with the lattice periodicity. In our theory, this phenomenon is described as an instability of the FL metallic phase within a random phase approximation scheme with a suitable effective electron–electron interaction [28,41,42], coming from high doping/high temperature. Of course, there is a connection with the physics of the strong correlated metal that exists at low doping (the evolution being gradual).

In the early times after their theoretical prediction, charge density waves were studied mainly through indirect probes due to the experimental difficulty of having a direct observation of this phenomenon [43,44]. An important step forward was made thanks to the improvement of the resonant X-ray scattering technique, which provided unambiguous direct evidence of charge density waves in all cuprate families [15,45–49]. According to the latest experimental results, charge density waves do not manifest themselves simply as a static modulation of the charge profile in the pseudogap phase but rather in the form of dynamical collective modes [15,49], as it had been theoretically predicted [41,42].

Below a certain critical temperature, dependent on the doping level, charge density waves may be able to establish long-range order, characterized by a finite wave vector lying on the CuO plane, which we will denote with $\mathbf{q}_c$. From soft X-ray scattering experiments, the presence of a narrow peak around $\mathbf{q}_c \approx (0.31, 0)$ r.l.u. (reciprocal lattice units) in the nearly-elastic portion of the spectrum is evident [15]. Of course, tetragonal symmetry of the lattice implies that similar peaks are observed at a star of wave vectors which are equivalent under the point group symmetries. Interesting phenomena involve charge density waves when an orthorhombic distortion occurs [27], but we shall concentrate on the tetragonal phase, although all our conclusions are essentially unchanged in the presence of a weak orthorhombic distortion.

Experiments indicate that the region of the phase diagram in which charge density waves are truly long-ranged is entirely located below the superconducting dome [50]. Approaching the QCP, the correlation length of the related collective excitations grows and they become able to mediate a stronger and stronger interaction between the fermion quasiparticles. In principle, this interaction can be singular enough to lead to violation of the FL properties, such as, e.g., the $T^2$ behavior of the resistivity at low temperature. The strange-metal phase of cuprates is characterized by linear-in-temperature resistivity which starts from very low temperatures and extends up to the highest measured

temperatures [8,9]. This behavior, which is one of the most evident violations of Landau's FL theory, indicates the lack of an intrinsic energy scale, namely a scale-invariant transport. Usually, if the constant of proportionality between the scattering rate and the temperature is $O(1)$ in natural units of $k_B/\hbar$, this behavior is referred to as Planckian behavior [51], but we prefer to use the term strange-metal behavior. Charge density waves cannot explain this specific violation of the FL behavior, because their ordering occurs at finite wave vector $\mathbf{q}_c$. This implies that low-temperature scattering can only occur between regions of the Fermi surface that are connected by $\mathbf{q}_c$, usually called hot spots. At the same time, as long as the charge density waves stay dynamical and the symmetry is not broken (i.e., one is still inside the FL phase), the hot spots are points where the scattering is strongest, but the Fermi surface is not spoiled [52]. Since the other regions of the Fermi surface are not involved in this scattering process, most of the quasiparticles retain their standard FL behavior [53]. It became evident that what was responsible for the strange-metal properties of cuprates had yet to be identified.

## 4. Charge Density Fluctuations

A major breakthrough in understanding the strange-metal behavior was achieved in 2019, performing resonant inelastic X-ray scattering experiments on $YBa_2Cu_3O_{7-\delta}$ and $Nd_{1+x}Ba_{2-x}Cu_3O_{7-\delta}$ films [14]. A clear peak in momentum space, centered at $\mathbf{q} = \mathbf{q}_c$, has been identified in the quasi-elastic component of the inelastic X-ray scattering spectra. At sufficiently high temperatures, the peak appears broad and can be fitted with a simple Lorentzian profile. At lower temperature, two Lorentzian profiles are needed to fit the observed peak: a broad one, which is very similar to the one observed at high temperature, and a narrow one, which presents all the characteristics previously observed in several compounds and commonly attributed to charge density waves. The broad peak was attributed to collective excitations, called charge density fluctuations, that pervade the phase diagram of cuprates. The centers of the two Lorentzian peaks are very close to each other, meaning that the characteristic wave vectors of the two collective excitations are very similar, suggesting that they may have a common origin. Charge density fluctuations can be described as a sort of aborted charge density waves which, for some reason (e.g., locally enhanced disorder or other slightly less favorable conditions), fail to develop long-range correlations.

It was immediately realized that the presence of the charge density fluctuations could provide the solution to the problem of the strange-metal behavior [12]. In fact, these collective excitations, having a small characteristic energy scale $\approx 10\,\mathrm{meV}$ and being so broad in momentum space, provide a microscopic scattering mechanism for the electron quasiparticles that is apt to affect all states on the Fermi surface nearly equally, resulting in a rather strong and essentially isotropic scattering rate. Moreover, unlike charge density waves, charge density fluctuations are present in a wide region of the phase diagram, encompassing the whole strange-metal phase. This strongly suggests that they may have an important role in the phenomenology of this phase.

However, the description of the strange-metal phase of cuprates through charge density fluctuations still has a problem. As we will clarify shortly, these collective excitations produce a linear-in-temperature resistivity at high temperature, but a $T^2$ FL behavior is recovered at low temperature. To suppress the temperature scale below which the FL behavior is recovered, a suitable mechanism must be invoked. We realized that the missing ingredient had to be identified in a dramatic increase in the Landau damping of the charge density fluctuations [13].

## 5. Role of the Landau Damping

Within the formalism of many body theory, collective excitations and fermion quasiparticles are described by the corresponding propagators. The form of the propagator for charge

density waves and charge density fluctuations has been deduced long ago [12,41,42,54] and takes the standard Ginzburg–Landau form in the Gaussian approximation,

$$\mathcal{D}(\mathbf{q}, \omega) = \frac{1}{m + \bar{v}|\mathbf{q} - \mathbf{q}_c|^2 - i\gamma\omega - \frac{\omega^2}{\overline{\Omega}}}, \tag{1}$$

which is a dynamical generalization of the Ornstein–Zernike correlator. Here, $m$ represents an energy scale (usually called the mass of the collective excitations) which measures the distance from criticality ($m \propto \xi^2$, where $\xi$ is the correlation length that diverges at criticality), $\bar{v}$ is an energy scale (if momenta are measured in r.l.u.), such that the correlation length is $\xi = \sqrt{\bar{v}/m}$, $\overline{\Omega}$ is a high energy cutoff scale (if the Planck constant $\hbar$ is set to one, as we shall do henceforth, frequency and energy scales coincide). A crucial role within our scheme is played by the imaginary term in the denominator, which represents the Landau damping. The dimensionless parameter $\gamma$ quantitatively describes the magnitude of the damping; it is inversely proportional to the decay rate of the collective modes, $\tau_{\mathbf{q}}^{-1} = (m + \bar{v}|\mathbf{q} - \mathbf{q}_c|^2)/\gamma$. We point out that an overdamped harmonic oscillator has two characteristic time scales: one is a time scale which is usually short and inversely proportional to the damping parameter, while the other is a larger time scale, which is directly proportional to the damping and rules the longer-time relaxation. The time scale to which we refer in our work is the latter.

Although the expression of $\mathcal{D}(\mathbf{q}, \omega)$ is formally the same for charge density waves and charge density fluctuations, these differ in the values of the parameters. It is important to note that the value of $\gamma$ does not affect the coherence length, so that the difference between charge density waves (with relatively long coherence lengths) and charge density fluctuations (with coherence lengths as short as few lattice spacings) is only ruled by the parameter $m/\bar{v}$. Since in this paper we are going to focus on the role of charge density fluctuations, the values of the parameters will be fixed accordingly.

Starting from resonant X-ray scattering experiments, it is possible to fit the values of $m$, $\bar{v}$ and $\overline{\Omega}$. Unfortunately, though these experiments confirm the presence of charge density fluctuations in $La_{1.6-x}Eu_{0.4}Sr_xCuO_4$ and $La_{1.6-x}Nd_{0.4}Sr_xCuO_4$ (we will carry out our analysis on these compounds, for which both resistivity and specific heat have been measured), detailed data are not available to extract the parameters we need. We will use the parameters fitted from the data related to $Nd_{1+x}Ba_{2-x}Cu_3O_{7-\delta}$, namely $m = 15\,\text{meV}$, $\bar{v} = 1.3\,\text{eV}/(\text{r.l.u.})^2$ and $\overline{\Omega} = 30\,\text{meV}$ [12], which are reasonable estimates.

Our proposal, which is based on recent experimental observations [9,11], provides the possibility for $\tau_{\mathbf{q}}^{-1}$ to become very small at every $\mathbf{q}$ because of a diverging $\gamma$ at fixed $\xi$, rather than becoming small only near $\mathbf{q}_c$ because of a diverging $\xi$ (this is the standard slowing-down of critical phenomena). We will not provide a microscopic model for this mechanism, but we will limit ourselves to the phenomenological assumption that $\gamma$ depends both on temperature and doping, in such a way as to diverge at a certain doping $p^*$ (slightly) larger than the doping $p_c$ for the QCP of charge density waves. We will show that this assumption is sufficient to explain two of the most peculiar aspects of the strange-metal phase: the aforementioned linear-in-temperature resistance and a divergent specific heat, which has been observed in several compounds. It is important to emphasize that in our scenario, $p^*$ should not be identified with $p_c$. The distance between the two points is not universal and changes in the various cuprate families. For our scenario to be meaningful and coherent, we only need the two points to not be very far apart. We need a mechanism that brings charge density fluctuations to be characterized by a sufficiently low energy scale, so as to play a relevant role as scatterers for the electron quasiparticles, and this occurs if these fluctuations are not too far from a quantum critical point for charge ordering. At the same time, since they must be broad enough in momentum space to mediate an isotopic scattering, they cannot be too close to $p_c$. A dynamical instability can occur at a doping $p* > p_c$, such that the characteristic time scale of the fluctuations grows very large and the linear resistivity extends over a wide temperature range, down to very low temperature. This is achieved, e.g., if the parameter $\gamma$ grows large.

The request for a large $\gamma$ is a consequence of the transport properties observed in cuprates: the temperature scale $T_{\text{FL}}$ above which the system loses its FL properties is proportional to $m/\gamma$; therefore, a large $\gamma$ would extend the non-FL behavior down to low temperatures. The functional form of $\gamma = \gamma(p, T)$ can be deduced from both the resistivity and the specific heat measurements. We require that $\gamma = \gamma(p, T)$ depends on $p$ only through the difference $|p - p^*|$, where $p^*$ denotes the doping level at which $\gamma$ may diverge, and that it increases as $p$ approaches $p^*$ at any fixed $T$. Moreover, we want $\gamma(p, T)$ to be decreasing function of $T$, at least for sufficiently low temperatures, and that it diverges at $p = p^*$, while it is regular elsewhere. The phenomenological expression we adopted for the damping parameter is

$$\gamma(p, T) = \left[ \frac{A}{\log(1 + T_0/\Theta_T)} + B|p - p^*| \right]^{-1},\qquad(2)$$

where $\Theta_T \equiv \min(T, \overline{T})$ for a temperature scale $\overline{T}$ above which the temperature dependence of $\gamma$ saturates. $A$, $B$ and $T_0$ can be taken as fit parameters for both resistivity and specific heat data. The temperature scale $\overline{T}$ cannot be fitted starting from low-temperature specific heat data [11]; we can only say that it must be at least equal to the highest temperature reached in low-temperature specific heat measurements.

## 6. Singular Behavior of Specific Heat

As we have already mentioned, several specific heat measurements, carried out in 2019 on some samples of $La_{1.6-x}Eu_{0.4}Sr_xCuO_4$ and $La_{1.6-x}Nd_{0.4}Sr_xCuO_4$, highlighted the presence of a thermodynamic singularity at the QCP, signaled by the fact that the ratio $C_V/T$ seems to diverge as $\log(1/T)$ at fixed $p = p^*$ [11]. In principle, such behavior could be due to fermion quasiparticles if their chemical potential approaches a van Hove singularity, which we know to be present close to the critical doping level. Nevertheless, the presence of disorder and the weak coupling between lattice planes strongly smoothen the singularity; hence, it was argued that the fermion quasiparticles alone are not able to account for the singularity.

Within our scheme, this anomalous behavior is rather due to the contribution of charge density fluctuations. Since the physical effect of an increasing $\gamma$ is to shift the spectral weight of charge density fluctuations down to lower and lower energies, we expect an increase in specific heat close to the critical doping, where $\gamma$ is large. This mechanism could then explain the behavior of specific heat around the critical doping level.

## 7. Methods

For the computation of the in-plane resistivity, we used a Boltzmann-equation approach, yielding [55]

$$\rho = \left[ \frac{e^2}{\pi^3 \hbar} \frac{2\pi}{d} \int \frac{k_F(\phi) v_F(\phi) \cos^2(\phi - \psi)}{\Gamma(\phi) \cos(\psi)} \, d\phi \right]^{-1},$$

where the $z$-axis lattice constant is taken as $d \approx 11$, which is approximately the known value for $La_{1.6-x}Nd_4Sr_xCuO_4$, while $k_F(\phi)$, $v_F(\phi)$ and $\Gamma(\phi)$, respectively, indicate the angular dependence of the Fermi momentum, Fermi velocity and scattering rate along the Fermi surface. Moreover, we have introduced

$$\psi = \arctan\left( \frac{1}{k_F} \frac{\partial k_F}{\partial \phi} \right).$$

Within our model, the scattering rate is given by the superposition of two distinct contributions: an isotropic scattering component due to quenched impurities (which we denote with $\Gamma_0$) and a scattering component due to charge density fluctuations, which is proportional to the imaginary part of the electron self-energy. We can therefore express $\Gamma(\phi)$

as $\Gamma_0 - \operatorname{Im}\Sigma(k_F(\phi), \omega = 0)$. We recall that the rate of the scattering mediated by charge density fluctuations depends very weakly on the angular variable, due to broadness of these collective modes in momentum space.

In order to compute the self-energy, we carried out a perturbative calculation up to the first relevant order. The analytic expression we obtain for the imaginary part of the self-energy is [56]

$$\operatorname{Im}\Sigma(\mathbf{k}, \omega) = -g^2 \int \frac{\mathrm{d}^2\mathbf{q}}{(2\pi)^2} \frac{(\omega - \varepsilon_{\mathbf{k}-\mathbf{q}})[b(\varepsilon_{\mathbf{k}-\mathbf{q}}) + f(\varepsilon_{\mathbf{k}-\mathbf{q}} - \omega)]}{[m + \bar{v}\,\eta_{\mathbf{q}} - (\omega - \varepsilon_{\mathbf{k}-\mathbf{q}})^2/\overline{\Omega}]^2 + (\omega - \varepsilon_{\mathbf{k}-\mathbf{q}})^2},\tag{3}$$

where $b(z) = [\mathrm{e}^{\beta z} - 1]^{-1}$ is the Bose function, $f(z) = [\mathrm{e}^{\beta z} + 1]^{-1}$ is the Fermi function, $g$ is the coupling constant which describes the interaction between fermion quasiparticles and collective modes, while $\eta_{\mathbf{q}} = 4 - 2\cos(q^{(x)} - q_c^{(x)}) - 2\cos(q^{(y)} - q_c^{(y)})$ is the lattice periodic dispersion of charge density fluctuations in momentum space.

Self-energy also plays a role in the computation of the specific heat. As a first step, since our idea is that the observed behavior of the specific heat is due to both electrons and collective modes, we subtract the electron contribution from specific heat data. Since the available specific heat data relate to the low temperature regime, we can estimate the fermion contribution by means of the Sommerfeld expansion:

$$\frac{C_V^f}{T} = \frac{\pi^2}{3}k_{\mathrm{B}}^2\,\rho_f(\omega = \mu),$$

where $\rho_f(\omega)$ is the fermion density of states, which takes into account also the self-energy correction, while $\mu$ is the chemical potential, which depends on the doping level. In order to compute the boson contribution to the specific heat, we start from the expression of the free energy in terms of the propagator and then evaluate the effect of a finite $\gamma$ by computing the excess of free energy $\delta\mathcal{F} = \mathcal{F}_\gamma - \mathcal{F}_{\gamma=0}$. We found the following expression:

$$\delta\mathcal{F} = \frac{1}{2\beta}\sum_{n,\mathbf{q}}\left[\log\left(\frac{\omega_n^2}{\overline{\Omega}} + \gamma|\omega_n| + \omega_{\mathbf{q}}\right) - \log\left(\frac{\omega_n^2}{\overline{\Omega}} + \omega_{\mathbf{q}}\right)\right],$$

where $\omega_{\mathbf{q}} = m + \bar{v}|\mathbf{q} - \mathbf{q}_c|^2$. Starting from this expression, we can compute the excess of internal energy $\delta\mathcal{U}$ using the thermodynamic identity $\delta\mathcal{U} = \frac{\partial}{\partial\beta}(\beta\,\delta\mathcal{F})$, which lead us to

$$\delta\mathcal{U} = k_{\mathrm{B}}T\sum_{n,\mathbf{q}}\left[\frac{\omega_{\mathbf{q}} + \frac{1}{2}\gamma|\omega_n|}{\frac{\omega_n^2}{\overline{\Omega}} + \gamma|\omega_n| + \omega_{\mathbf{q}}} - \frac{\omega_{\mathbf{q}}}{\frac{\omega_n^2}{\overline{\Omega}} + \omega_{\mathbf{q}}}\right].$$

This expression allows us to introduce the internal energy of collective modes as

$$\mathcal{U} = k_{\mathrm{B}}T\sum_n\sum_{\mathbf{q}}\frac{\omega_{\mathbf{q}} + \frac{1}{2}\gamma|\omega_n|}{\frac{\omega_n^2}{\overline{\Omega}} + \gamma|\omega_n| + \omega_{\mathbf{q}}}.$$

Note that this sum is formally divergent, so we will have to introduce a convergence factor, as it is customary in diagrams with closed loops. We carry out the Matsubara sum by means of residue theorem and we obtain the following expression for the thermal part of the internal energy

$$\mathcal{U} = \sum_{\mathbf{q}}\frac{1}{\pi}\int_0^\infty \frac{\gamma\omega\left(\frac{\omega^2}{\overline{\Omega}} + \omega_{\mathbf{q}}\right)}{\left(\frac{\omega^2}{\overline{\Omega}} - \omega_{\mathbf{q}}\right)^2 + \gamma^2\omega^2}\,b(\omega)\,\mathrm{d}\omega.$$

The specific heat is given by differentiating $\mathcal{U}$ with respect to $T$ and dividing by the number of the cells of the system $N$. We can formally express the boson contribution to the specific heat as

$$C_V^b = \frac{\partial}{\partial T} \int_0^\infty \rho_B(\omega)\, \omega\, b(\omega)\, d\omega,$$

where $\rho_B(\omega)$ plays the role of an effective density of states and is defined as

$$\rho_B(\omega) = \frac{1}{N} \sum_\mathbf{q} \frac{1}{\pi} \frac{\gamma\left(\dfrac{\omega^2}{\overline{\Omega}} + \omega_\mathbf{q}\right)}{\left(\dfrac{\omega^2}{\overline{\Omega}} - \omega_\mathbf{q}\right)^2 + \gamma^2\omega^2}.$$

We carry out the sum over $\mathbf{q}$ with the help of a suitable density of states, defined as

$$g(\varepsilon) = \frac{1}{N} \sum_\mathbf{q} \delta(\bar{v}|\mathbf{q} - \mathbf{q}_c|^2 - \varepsilon) \approx \frac{1}{\pi \bar{v}} \theta(\Lambda - \varepsilon). \tag{4}$$

Although the first Brillouin zone is three-dimensional, the dispersion is only on the $xy$ plane. In order to carry out the sum, we approximated the quarter of the Brillouin zone with a circle centered at each of the four equivalent $\mathbf{q}_c$ with radius $\bar{q}$, chosen in such a way that the area of this circle is $1/4$ of the area of the first Brillouin zone, namely $\bar{q} = \sqrt{\pi}$. This introduces an ultraviolet cutoff $\Lambda$, whose value is $\Lambda = \bar{v}\bar{q}^2 = \pi\bar{v}$. The full analytic expression for $\rho_B(\omega)$ is

$$\begin{aligned}
\rho_B(\omega) &= \frac{\gamma}{2\pi\Lambda} \log \frac{\left(m + \Lambda - \frac{\omega^2}{\overline{\Omega}}\right)^2 + \gamma^2\omega^2}{\left(m - \frac{\omega^2}{\overline{\Omega}}\right)^2 + \gamma^2\omega^2} \\
&+ \frac{2\omega}{\pi\Lambda\overline{\Omega}} \left[\arctan\left(\frac{m + \Lambda}{\gamma\omega} - \frac{\omega}{\gamma\overline{\Omega}}\right) - \arctan\left(\frac{m}{\gamma\omega} - \frac{\omega}{\gamma\overline{\Omega}}\right)\right].
\end{aligned} \tag{5}$$

For the computation of the specific heat, we are considering the low temperature limit, so it is reasonable to assume both $\omega \ll \sqrt{\overline{\Omega}(m + \Lambda)}$ and $\gamma\omega \ll m + \Lambda$. Within this regime, we can approximate $\rho_B(\omega)$ as

$$\rho_B(\omega) \approx \rho_B(\omega=0) = \frac{\gamma}{\pi\Lambda} \log\left(1 + \frac{\Lambda}{m}\right). \tag{6}$$

The lower the temperature, the more valid the approximation becomes. The expression of the specific heat we obtain in this regime is

$$\frac{C_V^b}{T} \approx \frac{\pi^2}{3} k_B^2 \rho_B(\omega=0) = k_B^2 \frac{\gamma}{3\bar{v}} \log\left(1 + \frac{\Lambda}{m}\right). \tag{7}$$

## 8. Effect of a Three-Dimensional Dispersion

Although cuprates can be well described as two-dimensional systems, we know that a weak coupling between lattice planes must exist, and it is partly responsible for the long-range correlation phenomena at finite temperature, otherwise forbidden by Mermin–Wagner theorem [57].

It may be interesting to evaluate the effect of a dispersion along the $z$-axis, which we will denote with $\bar{v}_\perp$. The dispersion becomes $\omega_\mathbf{q} = m + \bar{v}|\mathbf{q} - \mathbf{q}_c|^2_{xy} + \bar{v}_\perp q_z^2$. We can

compute $g(\varepsilon)$ in a formally identical way to that applied in Equation (4), but this time we obtain the following result:

$$\Lambda g(\varepsilon) = \begin{cases} \sqrt{\varepsilon/E} & \text{if } 0 < \varepsilon < E \\ 1 & \text{if } E < \varepsilon < \Lambda \\ 1 - \sqrt{(\varepsilon - \Lambda)/E} & \text{if } \Lambda < \varepsilon < E + \Lambda \\ 0 & \text{elsewhere,} \end{cases}$$

where we have introduced the parameter $E \equiv \pi^2 \bar{v}_\perp / d^2$. Of course, in order for this expression to make sense, the condition $E < \Lambda$ must hold, which can be expressed as $\bar{v}_\perp < \bar{v}_\perp^{\max}$, where $\bar{v}_\perp^{\max} \equiv \bar{v} d^2 / \pi$. Experimentally, we know that $\bar{v}_\perp \ll \bar{v}_\perp^{\max}$, but it may be interesting to evaluate the trend of the specific heat as the parameter varies within its entire validity domain, which is what we show in Figure 2.

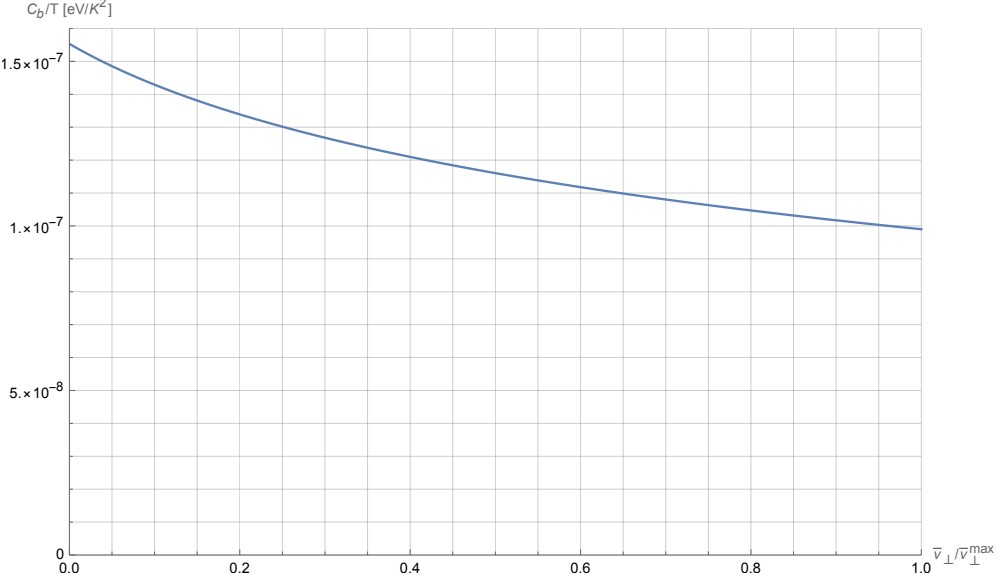

**Figure 2.** Plot of $C_V^b / T$ as a function of $\bar{v}_\perp / \bar{v}_\perp^{\max}$ for $0 < \bar{v}_\perp < \bar{v}_\perp^{\max}$. Parameter values are $m = 15\,\text{meV}$, $\bar{v} = 1.3\,\text{eV}/(\text{r.l.u.})^2$, $\gamma = 1$, $\overline{\Omega} = 30\,\text{meV}$ and $d/a = 11$.

The integral in $d\varepsilon$ which defines a closed form for $\rho_B(\omega)$ can be solved analytically, and we can obtain an expression similar to Equation (5), which tends to it in the limit $E \to 0$. Here, we limit ourselves to exhibiting only the approximate form $\rho_B(\omega) \approx \rho_B(\omega=0)$ for the density of states, valid in the low temperature limit

$$\rho_B(\omega=0) = \frac{\gamma}{\pi \Lambda} \left[ 2\sqrt{\frac{m+\Lambda}{E}} \arctan\left(\sqrt{\frac{E}{m+\Lambda}}\right) \right.$$
$$\left. -2\sqrt{\frac{m}{E}} \arctan\left(\sqrt{\frac{E}{m}}\right) + \log\left(1 + \frac{\Lambda}{m+E}\right) \right]$$

It is immediate to note that this new expression tends to the one in Equation (6) in the limit $E \to 0$, as expected. Of course, also within this approximation, a linear relation between $C_V^b / T$ and $\gamma$ is obtained, which is analogous to Equation (7).

As we can see from the plot in Figure 2, the effect of a dispersion along $z$-axis is to reduce specific heat. Since $\bar{v}_\perp \ll \bar{v}$, this effect is essentially negligible. In all the computations that follow, we will always consider $\bar{v}_\perp = 0$, while keeping in mind what effect this parameter has on the thermodynamic properties of the system.

### 9. Results and Conclusions

Our most relevant result is that an increasing (possibly diverging) $\gamma$ would be able to explain the shift of the non-FL behavior to lower and lower temperatures, without affecting the correlation length $\xi$. In fact, from the point of view of transport properties, the effect of $\gamma \neq 1$ is formally equivalent to the following rescaling: $m \to m/\gamma$, $\bar{v} \to \bar{v}/\gamma$, $\overline{\Omega} \to \gamma \overline{\Omega}$ and $g^2 \to g^2/\gamma$. Of course, the correlation length is not affected by this rescaling; this ensures that the scattering remains isotropic on the Fermi surface. This request would not be satisfied if $m$ and $\bar{v}$ were rescaled separately (the effect of change of the scale $\overline{\Omega}$ is negligible at low temperature). From Figure 3, it is evident that making $m$ and $\bar{v}$ vary separately does not help to extend the linear regime to lower temperatures, while Figure 4 clearly shows that an increase in $\gamma$ corresponds to an extension of the linear regime for resistivity.

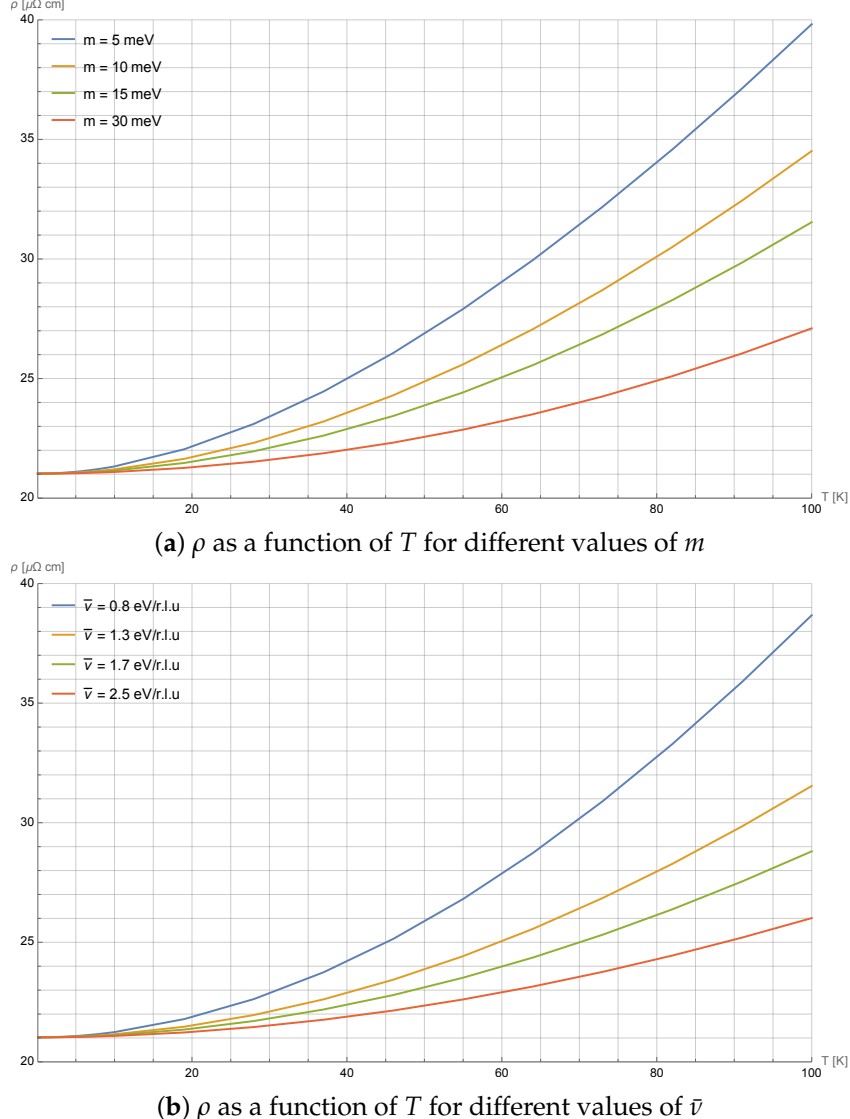

(**a**) $\rho$ as a function of $T$ for different values of $m$

(**b**) $\rho$ as a function of $T$ for different values of $\bar{v}$

**Figure 3.** Plot of the resistivity as a function of temperature. Parameter values, when set, are $m = 15\,\text{meV}$, $\bar{v} = 1.3\,\text{eV}/(\text{r.l.u.})^2$, $\gamma = 1$, $\overline{\Omega} = 30\,\text{meV}$, $g^2 = 0.0415$ and $\Gamma_0 = 12.3\,\text{meV}$. Note that the resistivity value at $T = 0\,\text{K}$ is greater than zero and it is the same for all curves, as this value is determined solely by $\Gamma_0$.

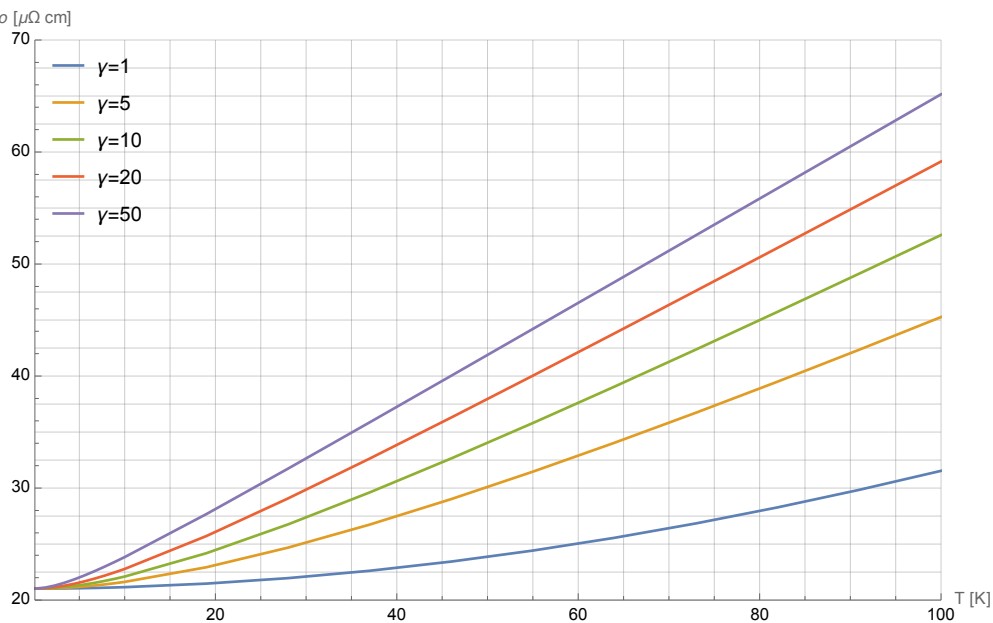

**Figure 4.** Plot of the resistivity as a function of temperature for different values of $\gamma$. Parameter values are $m =$ 15 meV, $\bar{v} = 1.3$ eV/(r.l.u.)$^2$, $\overline{\Omega} = 30$ meV, $g^2 = 0.0415$ and $\Gamma_0 = 12.3$ meV.

From resistivity data, there is no way to deduce a dependence of $\gamma$ on the temperature in the low temperature limit; however, we can deduce the order of magnitude of this parameter. We estimated that, in order to obtain a linear dependence down to less than 5 K, as required by La$_{1.6-x}$Nd$_{0.4}$Sr$_x$CuO$_4$ data, we need $\gamma$ to be of order 50 or 60 at low temperature. In order to fit the data relating to La$_{1.6-x}$Eu$_{0.4}$Sr$_x$CuO$_4$, it is sufficient that $\gamma$ is of order 20 or 30.

Another relevant finding is that overdamped charge density fluctuations give a contribution $C_V^b$ to the specific heat of the system, and this contribution grows large with increasing damping $\gamma$. In Figure 5, we show the dependence of ratio $C_V^b/T$ on the four parameters that appear in our theory, at fixed temperature of 1 K. From the plots, it is evident that the dependence on $\overline{\Omega}$ is completely negligible, as we had already mentioned, while the increase in the parameters $m$ and $\bar{v}$ causes a reduction in the specific heat. The most relevant dependence is that on $\gamma$. We expect that the ratio $C_V^b/T$ is generally an increasing function of $\gamma$. In particular, the dependence on $\gamma$ is linear in the low temperature limit, as can be seen from formula (7). In Figure 5c, we show $C_V^b/T$ as a function of $\gamma$ at fixed temperature $T = 1$ K, which is roughly a linear dependence, as we expect. In order to correctly reproduce the boson contribution to specific heat, we would need $\gamma$ to be of order 1. We know that this order of magnitude is too small to fit the resistivity curves observed experimentally. Consequently, a correct reproduction of resistivity data inevitably involves an overestimation of the specific heat. Our fit for the specific heat is able to capture only the qualitative trend of the observed curves, but this value is amplified uniformly over all temperatures. In particular, by consistently fitting the resistivity and specific heat data, we found $T_0 = 50$ K, $p_c = 0.235$, $A = 0.056$, $B = 0.87$ for La$_{1.6-x}$Nd$_{0.4}$Sr$_x$CuO$_4$ and $T_0 = 37$ K, $p_c = 0.232$, $A = 0.117$, $B = 2.84$ for La$_{1.6-x}$Eu$_{0.4}$Sr$_x$CuO$_4$, while resistivity data allowed us to find $g^2 = 0.045$ and $\Gamma_0 = 13.7$ meV for La$_{1.6-x}$Nd$_{0.4}$Sr$_x$CuO$_4$ and $g^2 = 0.0415$ and $\Gamma_0 = 12.3$ meV for La$_{1.6-x}$Eu$_{0.4}$Sr$_x$CuO$_4$. The resistivity trend has been reproduced correctly down to the lowest temperatures for which data are available. The overall amplification factor resulting from the specific heat fit is 30 for La$_{1.6-x}$Nd$_{0.4}$Sr$_x$CuO$_4$ and 11 for La$_{1.6-x}$Eu$_{0.4}$Sr$_x$CuO$_4$. We believe that a possible origin for this amplification may lie in an overestimation of the real number of collective charge density fluctuation degrees of freedom contributing to the specific heat. It may be possible that charge density fluctuations live, e.g., on a coarse-grained lattice with larger effective spacing [13].

In Figure 6, we report the trend of the specific heat as a function of the doping level for both $La_{1.6-x}Nd_{0.4}Sr_xCuO_4$ and 11 for $La_{1.6-x}Eu_{0.4}Sr_xCuO_4$ compounds at different temperatures.

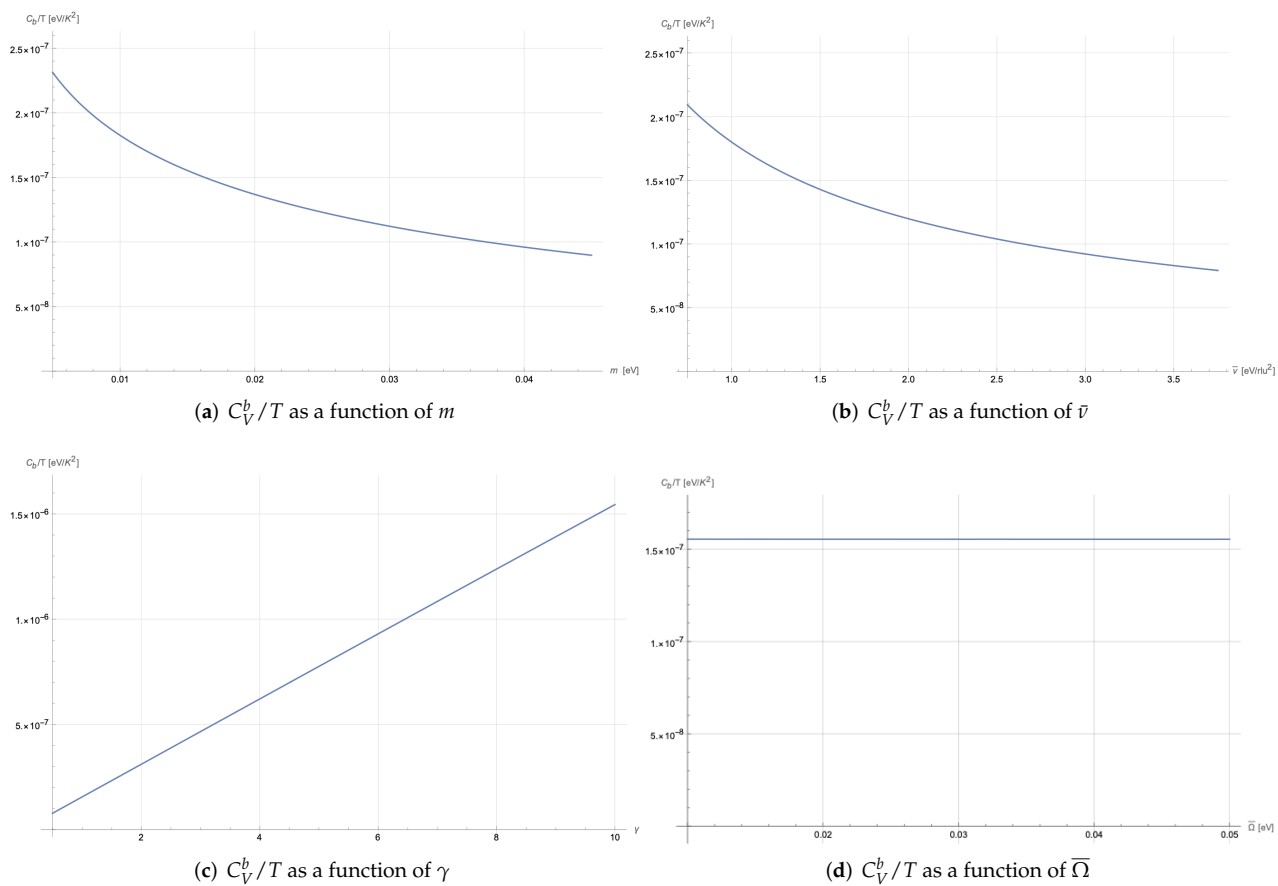

**(a)** $C_V^b/T$ as a function of $m$

**(b)** $C_V^b/T$ as a function of $\bar{v}$

**(c)** $C_V^b/T$ as a function of $\gamma$

**(d)** $C_V^b/T$ as a function of $\overline{\Omega}$

**Figure 5.** For each curve, we show the plot of the specific heat at $T = 1\,\text{K}$ when a single parameter changes, leaving all the others fixed. Parameter values, when set, are $m = 15\,\text{meV}$, $\bar{v} = 1.3\,\text{eV}/(\text{r.l.u.})^2$, $\gamma = 1$ and $\overline{\Omega} = 30\,\text{meV}$.

The two main findings discussed above show that, although our description lacks a microscopic scheme that can provide the dependence of $\gamma$ on temperature and doping level, our charge density fluctuations model is able to account for two highly anomalous behaviors of cuprates and that it is possible to do so by intervening only on a single term that appears within the theory. This would suggest that charge density fluctuations could play a central role in the phenomenology of the strange-metal phase of these systems. Moreover, an interesting aspect of our approach is that it shifts the role of the divergence of the correlation length to the divergence of the damping parameters of short-ranged fluctuations, thus introducing a new descriptive scheme for understanding non-FL behavior. This scheme could be applied not only in the description of cuprates but also of other materials in which a violation of FL behavior near QCPs has been observed, such as iron-based superconductors [58] or heavy-fermions metals [59].

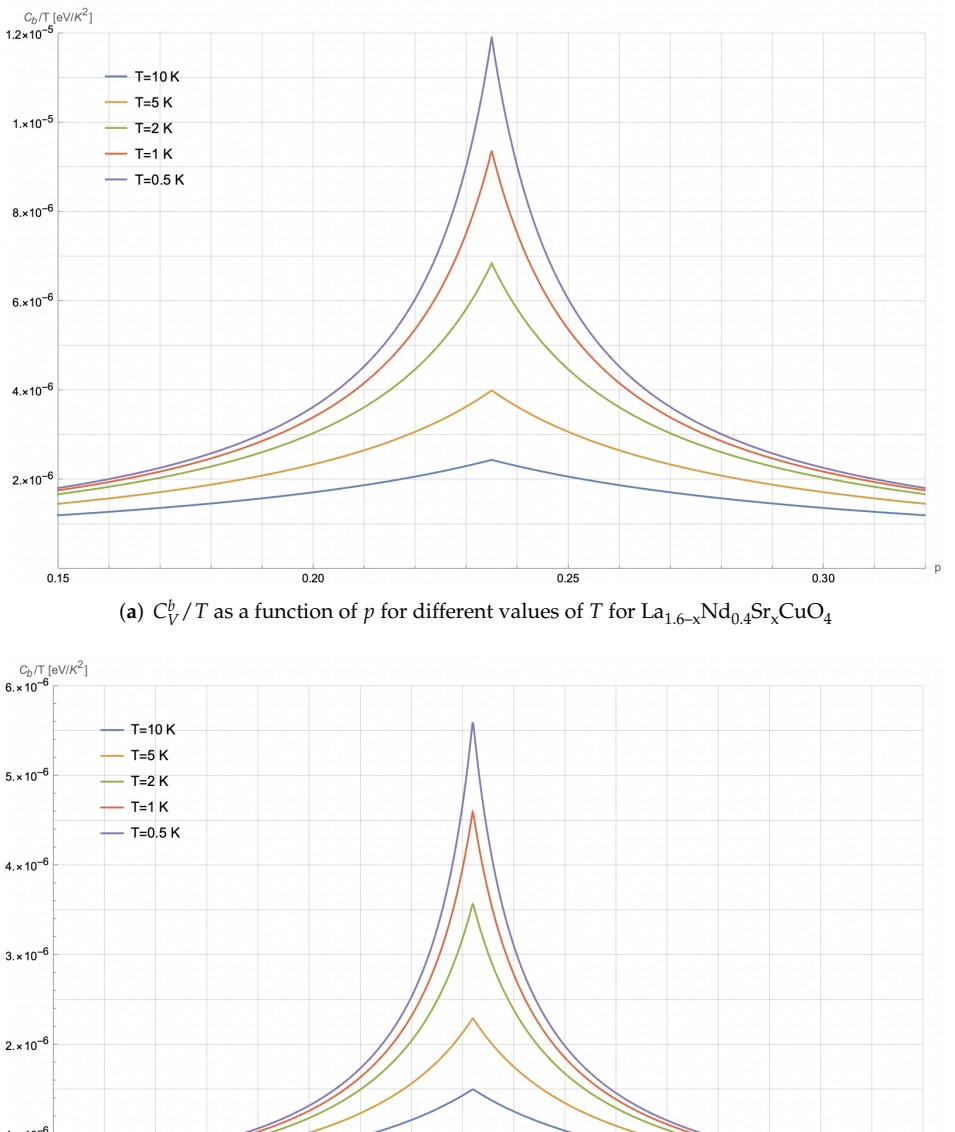

(**a**) $C_V^b/T$ as a function of $p$ for different values of $T$ for $La_{1.6-x}Nd_{0.4}Sr_xCuO_4$

(**b**) $C_V^b/T$ as a function of $p$ for different values of $T$ for $La_{1.6-x}Eu_{0.4}Sr_xCuO_4$

**Figure 6.** Plot of the specific heat as a function of the doping level for different temperatures. Parameter values are $m = 15\,\text{meV}$, $\bar{v} = 1.3\,\text{eV}/(\text{r.l.u.})^2$, $\overline{\Omega} = 30\,\text{meV}$.

**Author Contributions:** The project was conceived by S.C., C.D.C. and M.G.; the numerical calculations and the plots were made by G.M. with contributions from G.S., S.C., C.D.C. and M.G.; and the manuscript was written by G.M. and S.C. with contributions and suggestions from G.S., C.D.C. and M.G. All authors have read and agreed to the published version of the manuscript.

**Funding:** This research was funded by "La Sapienza" University of Rome, through the projects Ateneo 2018 (Grant No. RM11816431DBA5AF), Ateneo 2019 (Grant No. RM11916B56802AFE) and Ateneo 2020 (Grant No. RM120172A8CC7CC7), from the Italian Ministero dell'Università e della Ricerca, through the Project No. PRIN 2017Z8TS5B, and also by the Deutsche Forschungsgemeinschaft under SE806/19-1.

**Institutional Review Board Statement:** Not applicable.

**Informed Consent Statement:** Not applicable.

**Data Availability Statement:** Data sharing not applicable to this article as no datasets were generated or analyzed during the current purely theoretical work. The theoretical analysis was carried out with C++ and FORTRAN codes to implement various required numerical computations; the used codes are available from the corresponding authors on reasonable request.

**Acknowledgments:** We want to express our gratitude to Riccardo Arpaia, Antonio Bianconi, Lucio Braicovich, Claudio Castellani and Giacomo Ghiringhelli for stimulating discussions.

**Conflicts of Interest:** The authors declare no conflict of interest.

## Abbreviations

The following abbreviations are used in this manuscript:

| | |
|---|---|
| FL | Fermi Liquid |
| QCP | Quantum Critical Point |
| r.l.u. | reciprocal lattice units |

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
