# Peer review of "The Strange-Metal Behavior of Cuprates"

_condensedmatter, doi:10.3390/condmat7010029_

Round 1

Reviewer 1 Report

This is a lucidly written review of a theory of the strange metal behavior in cuprates based on charge fluctuations, as developed to a large extent over the past decades by the authors. A new aspect is a discussion of the contribution of critical charge fluctuations to the specific heat. Based on a few assumptions the authors present a coherent
picture.

The paper is generally well written. I am unhappy only about one major point, namely how magnetic fluctuations are discarded as an important player in cuprates near optimal doping, in particular in the first paragraph of Sec. 3. The authors argue that antiferrromagnetism is not important near optimal doping since ''antiferromagnetism is rapidly spoiled upon doping''. This is true only for Neel-type long-range order. Magnetic correlations extend to much higher doping, and they are a plausibel mechanism for pairing and for pseudogap behavior, as has been established in numerous controlled calculations. Moreover, in La2-xSrxCuO4 even frozen spins have been observed
upon suppression of superconductivity up to 19 percent doping. Hence, to be fair, the authors should at least acknowledge that magnetic fluctuations may be very important as well.

As a minor point it seems to me that on the left hand side of the equation in line 198 there should be the lifetime, not its inverse (the decay rate).

In summary, I recommend publication after the above points have been taken into account.

Reviewer 2 Report

This is a  nice work. Particularly the motivation and introduction are nicely chalked out. Before delving into the details of the work, I have some general remarks. One thing that confuses me throughout the paper is if the theoretical model is developed for some particular hole-doped copper oxides or for all of them. The reason I am confused is because the fundamental assumption of the work is related to the argument that pc for optimal Tc, p* across which linear in T behavior changes and pseudogap end point at T=0 are the same point. While there are at length discussions for several cuprates where this is most likely not that case, making a tri-critical theory for the 'critical' point redundant. I can still presume that there are some hole doped oxides where the theory is still relevant, but it is most likely not a general premise. So I would like to see a proper discussion and clarification around this point. 

Now coming to the details of the work, the authors argue based on some recent RIXS measurements that there is a second sub-lorentzian which is apparent similarities with may systems where charge-instabilities are the primary mechanism, that some hole doped cuprates can also have the charge density fluctuations as the main possible mechanism for the criticality. They build a damped density fluctuation phenomenological model to make a case for two main effects of criticality; linear in T resistivity and specific heat divergence at p*.   For them, the damping parameter increasing criticaly can explain the Fermi liquid scale to T=0, making it a potent phenomenological model for the criticality. 

In spirit, the work is sound and the arguments are  well presented   and the work should be published. However, I have some observations. In contrast with many trivial density fluctuation materials,  in hole doped copper oxides AFM phase is there and doping, in all likelihood, suppresses the long range AFM into incommensurate magnon fluctuations and it is very likely that some dynamic incommensurate magnon fluctuations survive along with dynamic density fluctuations for all dopings p~p*. What would be the consequence of that on the theoretical model authors have, particularly on the fact that damping parameter can not account for the correct estimates of linearity in resistivity and divergence in susceptibility in it's present pure 'charge' form? 

Secondly, the phenomenological model is more suited for approaching p* from below, rather that from above where the Fermi liquid scale is much higher. I would like to know authors' views on that?

Further, the paper is written in a way that SC is suppressed by some magnetic fields and that unveils the underlying criticality. At such low energy scales, magnetic field not only suppresses the SC, but can also lead to weak changes to the Fermiology which might be sufficient for removal of nesting. Is there a clear reason to believe, that in such studies field is not a component of the relevant physics?

Finally, somewhere in the paper the authors talk about tetragonal phases of hole doped cuprates. The T* line hits the tetragonal-orthorhombic phase transition temperature at p=0. Would the authors think of the relevant phonons at finite dopings as an extension of the same phonons at p=0 that drives the structural transition (in the sense of usual perovskites)? So what holds the tetragonal phase at finite dopings down to low temperatures? If those phonons are no way responsible for the dynamic density fluctuations, would not the pseudogap line be vertical, instead of being tilted?   

Reviewer 3 Report

The work " The strange-metal behavior of cuprates" is suitable for publication in the journal "Condensed Materials".

The work is very interesting since the problem of the strange metallic behavior of superconductors remains open at present.

But there are a few comments:

  1. In the abstract, the authors say "fluctuations provide the mechanism of microscopic scattering", but there is little mention of this in the text. What causes scattering and how does long-range order stop? I propose to bring the idea on what it is based on for experiments.
  2. Secondly, you say: “Charge density waves cannot explain this specific violation of the FL behavior, since their ordering occurs at a finite wave vector qc. This means that low-temperature scattering can only occur between regions of the Fermi surface connected by qc (commonly called hot spots) classically they are called singular points, "hot" spots were introduced later How is the Fermi velocity (v_f = ?) determined if it does not have a derivative at these points (e_k = 0 is not a Fermi surface)?
  3. I suggest that the authors add a conclusion to focus on the main findings.

Round 2

Reviewer 1 Report

My suggestions have been properly addressed. I recommend publication in the present revised form.

Reviewer 2 Report

All the scientifically relevant questions are addressed sufficiently and to their merits. I strongly recommend the work for publication.